# Facile Construction of Iron/Nickel Phosphide Nanocrystals Anchored on N-B-Doped Carbon-Based Composites with Advanced Catalytic Capacity for 4-Nitrophenol and Methylene Blue

**DOI:** 10.3390/ijms23158408

**Published:** 2022-07-29

**Authors:** Cheng Pan, Guangying Yang, Haitao Yang, Feifan Wu, Lei Wang, Jungang Jiang, Yifan Zhang, Junxia Yuan

**Affiliations:** Hubei Provincial Key Laboratory of Green Materials for Light Industry, Hubei University of Technology, Wuhan 430068, China; gyingygy@163.com (G.Y.); pphtyang1979@aliyun.com (H.Y.); wufeifan98@163.com (F.W.); wanglei@hbut.edu.cn (L.W.); jungang.jiang@hbut.edu.cn (J.J.); zhangyifan@hbut.edu.cn (Y.Z.); shirley_yjx@163.com (J.Y.)

**Keywords:** biomass carrier, bimetallic phosphide, catalytic hydrogenation

## Abstract

The search for a simple and effective method to remove organic dyes and color intermediates that threaten human safety from the water environment is urgent. Herein, we report a simple method for constructing iron/nickel phosphide nanocrystals anchored on N-B-doped carbon-based composites, using steam-exploded poplar (SEP) and graphene oxide (GO) as a carrier. The stability and catalytic activity of N-B-Ni_x_Fe_y_P/SEP and GO were achieved by thermal conversion in a N_2_ atmosphere and modifying the Fe/Ni ratio in gel precursors. N-B-Ni_7_Fe_3_P/SEP was employed for the catalytic hydrogenation of 4-nitrophenol (4-NP) and methylene blue (MB), using sodium borohydride in aqueous media at room temperature. This showed much better catalytic performances in terms of reaction rate constant (0.016 S^−1^ and 0.041 S^−1^, respectively) and the activity factor, K (1.6 S^−1^·g^−1^ and 8.2 S^−1^·g^−1^, respectively) compared to the GO carrier (0.0053 S^−1^ and 0.035 S^−1^ for 4-NP and MB, respectively). The strong interaction between the carrier’s morphology and structure, and the vertically grown bimetallic phosphide nanoclusters on its surface, enhances charge transfer, electron transfer kinetics at the interface and Ni-Fe phosphide dispersion on the nanoclusters, and prevents dissolution of the nanoparticles during catalysis, thereby improving stability and achieving catalysis durability. These findings provide a green and simple route to efficient catalyst preparation and provide guidance for the rational selection of catalyst carriers.

## 1. Introduction

Global water pollution is getting more severe, particularly regarding organic dyes and colour intermediates, endangering human safety. Nitrophenols are a common class of highly toxic, non-biodegradable organic pollutants, discovered in wastewater from the dyeing, pesticide and pharmaceutical industries. They are toxic to aquatic organisms and cause a variety of human health issues, such as injury to the liver, kidneys, blood and central nervous system [1,2,3]. Catalytic hydrogenation is one of the most attractive options for the removal of organic dyes and dye intermediates from the aqueous environment. The hydrogenation result, such as aminophenol, is less poisonous and more readily calcified than nitrophenol. More importantly, aminophenols can be employed as chemical intermediates in the synthesis of drugs, dyes and corrosion. As a result, the conversion of nitrophenols to value-added aminophenols is of great significance in both environmental remediation and industrial synthesis. Due to its great effectiveness and ease of application, sodium borohydride (NaBH_4_) reduction is regarded as one of the most effective procedures for the elimination of aromatic dyes and nitro compounds. The reduction reaction, on the other hand, become faster only in the presence of precious metal catalysts.

Nickel and noble metals are commonly used catalysts for the catalytic hydrogenation of 4-NP. Nickel is a transition metal, which is more abundant in nature, so the cost of nickel catalysts is low. Furthermore, nickel has a high catalytic activity in oxidation and hydrogen release processes. Supported nickel-based catalysts are widely utilised for the catalytic hydrogenation reactions of numerous compounds due to their low cost and high activity. Current research hotspots have discovered that synergistic interactions between several different metals can greatly improve the catalytic performance of bimetallic nanoparticles, since bimetallisation enhances the properties associated with the presence of two separate metals. For example, Fe/Ni bimetallic particles have a high potential for improving the catalytic reducing activity, and Fe acts as a reducing agent in Fe-based bimetallic nanoparticle systems [4]. However, the discovery that nanocatalysts clump and are difficult to recycle restricts their industrial applicability.

Currently, the use of carriers such as polymer matrices, metal–organic skeletons and carbon-based materials to immobilise metal particles, is an efficient way to solve this issue [5]. Catalyst carriers can prevent nanoparticle aggregation and increase nanoparticle stability and catalytic effectiveness. The optimal catalyst material should have multi-scale pores and a layered structure to offer a large number of active sites, while also allowing for facile diffusion of chemicals to and from the active sites [6].

Compared with other carriers, carbon materials are widely employed for the preparation of different types of catalysts due to their high specific surface area, rich pore structure and excellent stability, such as biochar carriers were used to improve the reduction activity of nickel/iron nanoparticles against 1,1,1-trichloroethane [7]; Sludge was used as a catalyst for catalytic reduction of nitrite by pyrolysis [8]; Three-dimensional nitrogen-doped graphene was used to prepare hydrothermal catalysts for the reduction of 4-nitrophenol [9]; hydrothermal silver nanoparticles embedded in magnetic GO were used to prepare composites for the reduction of 4-nitrophenol (4-NP) and methylene blue (MB) [2], and so on.

However, the preparation of these carbon-based catalysts is difficult, time-consuming, inefficient, energy-consuming, and accompanied with toxic byproducts. As a result, it is critical to develop a cost-effective and efficient, carbon-based catalyst. As lignocellulosic materials are renewable and high in hydroxyl groups, they are an excellent carbon source for the production of carbon carriers. We present the use of steam-blasted pretreated poplar as a carrier source for carbon-based materials, which is used as an energy crop for biomass or biofuels due to its rapid growth and great potential for carbon reduction, and is classified as an essential industrial feedstock in many countries around the world. Due to its huge surface area and adjustable surface chemistry, GO has shown significant potential as a substrate for different catalysts and sensors as an emergent class of carbon-based material [2]. Meanwhile, it was revealed that the catalytic activity of nickel-loaded catalysts was improved by doping with transition metal ions (Ni, Cu, Fe, Cr) and active non-metal ions (C, N, S, F) [2,10]. Transition metal phosphides are a class of nanostructure-doped active materials, with good electrical conductivity, a rich phase structure and redox chemistry, and low cost [11].

To obtain high-performance hydrogenation catalysts for the removal of organic dyes and dye intermediates by a green and simple preparation method, the effects of various carrier-loaded bimetallic phosphide ratios on the stability and catalytic activity of a NaBH_4_-catalysed model hydrogenation of 4-NP and MB, were examined. The morphology and structure of the resulting catalysts were characterised in detail. Due to the so-called synergistic effect, bimetallic catalysts demonstrated higher catalytic activity than monometallic equivalents. The choice of carrier material is also vital in the reduction reaction, since the reactivity and stability of the bimetallic catalysts strongly depend on the properties of the carrier surface. In addition, the structural interaction between the carrier and metal particles was studied in contrast to the catalysts made with GO as the carbon carrier.

## 2. Results and Discussion

The FT-IR analyzer was used to detect the functional groups on the N-B-Ni_7_Fe_3_P/SEP and N-B-Ni_7_Fe_3_P/GO, as shown in Figure 1. The FT-IR spectra revealed substantial carbon carrier absorption at 3421, 1591 and 1363 cm^−1^, which is attributable to the stretching of O-H, C=C, and C-OH, respectively [12]. The peaks at 2833 cm^−1^ were associated with the C-H stretching vibration methyl groups. In contrast, the weak peaks at 1112 cm^−1^ are characteristic of the C-N stretching mode. The bands at 777 and 619 cm^−1^ were the characteristic vibrations for Metal-O (M-O) [13], thereby again demonstrating that Ni-Fe formed on SEP and GO. The absorption peaks in N-B-Ni_7_Fe_3_P/SEP-1, N-B-Ni_7_Fe_3_P/GO-1, N-B-Ni_7_Fe_3_P/SEP-2, and N-B-Ni_7_Fe_3_P/GO-2 at 777 and 619 cm^−1^ did not decline, showing that the catalytic process did not affect the transition metal particles on the carrier surface.

The morphology and microstructure information of N-B-Ni_7_Fe_3_P/SEP and N-B-Ni_7_Fe_3_P/GO was systematically investigated by electron microscope. Vertical Ni-Fe nanowire development at room temperature has little effect on the appearance of the structure of SEP and rGO carriers (inset of Figure 2a,b). Scanning electron microscope images (Figure 2a,b) depict that the interconnected ultra-thin nanowires without agglomeration tend to form 3D honeycomb structures, and the vertically arranged nanoflakes have interconnected macroporous structures that do not hinder the potential microporous structures. This intriguing structure is advantageous to the catalytic process because it exposes a large number of catalytic active centres and allows for electrons to travel quickly along vertical nanoflakes. The elemental composition and distribution of N-B-Ni_7_Fe_3_P/SEP and N-B-Ni_7_Fe_3_P/GO samples were further characterised by energy spectrum surface scanning (Figure 2c,d). The results demonstrated that Ni, Fe, P, B and N were equally distributed throughout the sample. Ni/Fe atomic ratios were determined to be 62:38 and 56:44, respectively.

To check the dispersion and size of Ni-Fe nanoparticles in the vertical growth in SEP and rGO carriers, TEM measurements were carried out. Figure 3a and c indicate that nanoparticles are consistently distributed in the nanosheets with no agglomeration, demonstrating that SEP and rGO sheets aid in the aggregation of Ni-Fe nanoparticles. After high-temperature phosphating treatment, the 3D honeycomb network structure was well maintained, which reveals that the 3D network structure had strong structural stability. The medium magnification TEM picture (Figure 3a,c) indicated that the nanoparticles had a homogeneous distribution and were anchored on the nanowires. The results suggested that Ni-Fe nanoparticles had a robust interaction with SEP and rGO carriers. The HRTEM image (Figure 3a,c) exhibited obvious lattice stripes, and the lattice spacing was 0.22 nm, corresponding to the crystal plane of bimetallic phosphate. The SAED pattern depicted the FCC properties of Ni-Fe nanoparticles (Figure 3a,c), suggesting that the nanoparticles were manufactured with polycrystalline features. Furthermore, the findings of energy-dispersive X-ray mapping (Figure 3b,d) revealed that the elements Fe, Ni, and P were equally distributed across the whole carbon, indicating the synthesis of bimetallic phosphides.

To determine the crystal phase of the prepared bimetallic phosphide, X-ray diffractometry (XRD), using an X-ray diffractometer, was also performed. As illustrated in Figure 4, the diffraction peaks acquired from the catalyst samples before and after usage may be indexed with the separated diffraction peaks in Ni_2_P (PDF No. 03-0953), Fe_2_P (PDF No. 51-0943) and FeP (PDF No. 65-2595), respectively. The diffraction pattern for PC had a broad peak at 21.3°, which was a characteristic of the (002) plane of graphitic carbon. As the substrate for supporting Fe\Ni\P nanoparticles, porous graphite carbon not only enhances the uniform dispersion of active sites, but also promotes charge/mass transfer. There were no apparent diffraction peaks in metal Fe and Ni in the XRD spectra of N-B-Ni_7_Fe_3_P/SEP and N-B-Ni_7_Fe_3_P/GO, indicating that Fe and Ni nanoparticles were well disseminated over SEP and GO, which was consistent with the TEM data.

The XPS survey scan spectrum (Figure 5I) confirmed that Ni, Fe, B, P, O, and C elements were present in the samples, conducting to character the chemical valence states of the composites. The Ni 2p1/2 and Ni 2p3/2 XPS (Figure 5II) of the N-B-Ni_7_Fe_3_P/SEP-1, N-B-Ni_7_Fe_3_P/GO-1, post-catalytic N-B-Ni_7_Fe_3_P/SEP-2 and N-B-Ni_7_Fe_3_P/GO-2 are observed at 874.5 and 856.4 eV, with additional intense satellite peaks characteristic of Ni^2+^ (862.6 and 880.6 eV, respectively) [14]. A new peak arises at 853.7 eV in N-B-Ni_7_Fe_3_P/SEP-1, showing that the electronic structure has altered during the phase transition following phosphating treatment, and there are some positively charged Ni species with Ni-P bonds. The XPS spectrum shows that the structure of the catalyst did not alter much before and after the catalytic reaction, indicating that the catalyst’s stability is good.

The peaks at 711.5 eV and 724.6 eV were assigned to Fe 2p3/2 and Fe 2p1/2 of Fe^2+^, whereas those at 715.4 eV and 728.1 eV correspond to Fe 2p3/2 and Fe 2p1/2 of Fe^3+^ [15]. The existence of Fe^2+^ and Fe^3+^ stems from the partially oxidised surface of N-B-Ni_7_Fe_3_P/SEP-1 and N-B-Ni_7_Fe_3_P/GO-1 during the storage and testing process. It is worth noting that the peak at 728.1 eV vanished during the reaction, showing that Fe3+ was engaged in the catalytic hydrogenation of 4-NP. These visible alterations are mostly due to the synergistic action of a range of metal components, which results in the effect of electron transfer in the hybrid.

As shown in Figure 5IV, the three components of the C 1 s spectrum (284.8, 286.3 and 289.0 eV) were attributed to C-C, C-O and O-C = O groups, respectively, which offer the sites that anchor the metal ions. It has been discovered that the intensity of C 1 s peaks at 286.3 eV (for C-O) is significantly higher in the N-B-Ni_7_Fe_3_P/GO (Figure 5(IVc)) catalyst than in N-B-Ni_7_Fe_3_P/SEP (Figure 5(IVa)) because of the oxygenated functional group. The intensity is effectively lowered in comparison to the sample following catalytic reaction (Figure 5(IVb,d)) due to the reduction process [16].

The high-resolution XPS spectra of P 2p (Figure 5V) can be deconvoluted into four peaks at 129.6, 130.5, 133.3, and 134.1 eV, being assigned to the featured P 2p3/2 and P 2p1/2 peaks for metal phosphides, and formation of P-O, and P = O bonds, respectively [17]. The P = O in this equation should come from oxidised P species. This is caused by common metal phosphides being exposed to air. Following the catalytic reaction, the recovered catalyst’s characteristic peak shifts to the negative direction, and its binding energy is lower than that of the element P0 (130.0 eV), indicating a strong bonding interaction between metal particles and partially negatively charged P- in the catalytic reaction [15]. These results show that there is a strong electronic coupling between Ni, Fe and P sites, and the electrons may transfer from Fe or P sites to Ni sites in the catalyst [18].

Further N_2_ sorption tests for N-B-Ni_7_Fe_3_P/SEP and N-B-Ni_7_Fe_3_P/GO (Figure 6a) depict the obvious type-IV plot, demonstrating that they possess a mesoporous structure. The isotherms (Figure 6) of the mixed oxides were categorised as type-IV with an H_3_ hysteresis loop, implying the presence of mesoporous materials with incision-like pore structure. The specific surface areas of N-B-Ni_7_Fe_3_P/SEP and N-B-Ni_7_Fe_3_P/GO were calculated to be 47.80 and 31.65 m^2^/g, respectively. The average pore size of N-B-Ni_7_Fe_3_P/SEP was about 1.44 nm, while those of N-B-Ni_7_Fe_3_P/GO was around 1.39 nm. The small-sized mesopores should form as a result of the nanoclusters developing vertically on the surface of the carbon carrier, and this structure helps to expose more active sites, boosting the catalyst’s overall activity.

The thermal stability of the carbon-based–supported Ni–Fe bimetallic phosphide catalyst was observed by thermogravimetric analysis. The TGA curves of the samples (Figure 7b) showed a maximum weight loss rate of less than 200 °C, which was attributed to the elimination of physisorbed water molecules. The final two steps in the weight loss rate of N-B-Ni_7_Fe_3_P/SEP-2 and N-B-Ni_7_Fe_3_P/GO-2 take place at 250–500 °C, corresponding to the decomposition of the light-volatile compound after catalytic reactions. The total weight loss of N-B-Ni_7_Fe_3_P/SEP-1, N-B-Ni_7_Fe_3_P/GO-1, post-catalytic N-B-Ni_7_Fe_3_P/SEP-2, and N-B-Ni_7_Fe_3_P/GO-2 in the range of 20–800 °C was 17.8%, 3.8%, 41.9% and 8.4%, respectively. These findings support the critical function of carbon-based carriers in the reduction, growth and stability of Ni-Fe bimetallic phosphides. Based on these findings, the Ni-Fe bimetallic phosphide catalyst supported on carbon, is thermally stable at temperatures as high as 800 °C.

It is known that 4-nitrophenol and methylene blue are typical, representative, hazardous dye intermediates and organic dyes known to the public [19]. Therefore, the catalytic hydroreduction in 4-NP and MB in the presence of excess NaBH_4_ at room temperature has been widely utilised as a reference system for assessing the catalytic activity of the resulting composites, which is of great importance for environmental protection [20].

Normally, the 4-NP solution demonstrates maximum absorption at 317 nm. However, after adding NaBH_4_ solution, the absorption peak in 4-NP immediately shifts from 317 to 400 nm, which is due to the alkaline conditions [21,22]. The dark yellow of the 4-NP solution rapidly vanishes (inset) in the presence of N-B-Ni_7_Fe_3_P/SEP, as shown in Figure 8a, indicating a rapid drop in the intensity of 400 nm absorbance. Simultaneously, a new peak formed at 300 nm and grew with reduction time, which was ascribed to 4-AP, confirming that 4-NP was reduced to 4-AP. The reduction in 4-NP with N-B-Ni_x_Fe_y_P/SEP (x:y = 5:0; 5:5; 7:3; 9:1) was completed within 1.5–15 min over 10 mg, respectively (Figure 8b), and N-B-Ni_7_Fe_3_P/GO (10 mg) were completed within 1.5 min and 10 min, respectively (Figure 8d). The quick conversion of 4-NP demonstrates the excellent catalytic capacity of N-B-Ni_7_Fe_3_P/SEP.

Considering that the reductant concentration is much higher than that of 4-NP (C_NaBH4_/C_4-NP_ = 100) in the reaction mixture, the pseudo-first-order rate constant (k, S^−1^) can be calculated through the formula: −ln(C_t_/C_0_) = kt, where t is the reaction time. C_0_ and C_t_ are the 4-NP concentrations at time 0 and t, respectively. As expected, a good linear correlation of ln(C_t_/C_0_) vs. reaction time t, was obtained (Figure 8c,f), and the kinetic rate constant k was estimated as 0.007 (R^2^ = 0.91), 0.013 (R^2^ = 0.90), 0.016 (R^2^ = 0.90), 0.0015 (R^2^ = 0.96) and 0.0053 (R^2^ = 0.86) S^−1^ for N-B-Ni_x_Fe_y_P/SEP (x:y = 5:0; 5:5; 7:3; 9:1) and N-B-Ni_7_Fe_3_P/GO, respectively. We computed the ratio of the rate constant K over the total weight of the nickel catalyst to compare various catalysts, where K = k/m. As a result, the activity factor K was calculated as 0.7, 1.3, 1.6, 0.15 and 0.53 S^−1^·g^−1^ for N-B-Ni_x_Fe_y_P/SEP (x:y = 5:0; 5:5; 7:3; 9:1) and N-B-Ni_7_Fe_3_P/GO, respectively. When compared to other precious metal catalysts, such as Ru/C (0.034 min^−1^) and Ru/PC-IM (0.198 min^−1^) [23], it is obvious that N-B-Ni_7_Fe_3_P/SEP has the highest activity factor.

Furthermore, the catalyst may be readily extracted from the reaction mixture and reused by employing an external magnet. The kinetic rate constant k was calculated as 0.007 (R^2^ = 0.98) and 0.0012 (R^2^ = 0.83) S^−1^ for N-B-Ni_7_Fe_3_P/SEP-2 and N-B-Ni_7_Fe_3_P/GO-2, respectively (Figure 8d,f).

Electron transfer is an important process for the reduction conversion of 4-NP. According to XPS investigations, the electronic interaction and transfer within the NiFe_2_P nanocrystals and with the SEP/GO matrix may aid in the adsorption and activation of 4-NP molecules. In the reaction process, the catalyst interacts with BH_4_^−^ to form surface bound hydride that subsequently attacks the adsorbed 4-NP, during which electron transfer from BH_4_^−^ (electron donor) to 4-NP (electron acceptor) occurs [2,8,11,24].

UV-Vis spectroscopy was also used to monitor the catalytic reaction process of MB [25]. The corresponding UV-Vis spectra are shown in Figure 9a,b. When the catalyst was added to the cuvette containing the mixture of MB and NaBH_4_ solution, the characteristic adsorption peak in MB rapidly decayed and almost disappeared after 1 min. The kinetic information of MB catalytic reduction could be obtained by ln(C_t_/C_0_) versus time t, as shown in Figure 9c, and the kinetic rate constant k was estimated as 0.041 (R^2^ = 0.91) and 0.035 (R^2^ = 0.91) S^−1^ for N-B-Ni_7_Fe_3_P/SEP and N-B-Ni_7_Fe_3_P/GO, respectively, which exceed the Au/3D-graphene (0.0238 min^−1^) [26] and Pd/3D-graphene (0.0689 min^−1^) [27]. The results further convincingly confirmed the excellent catalytic performance of the prepared N-B-Ni_7_Fe_3_P/SEP and N-B-Ni_7_Fe_3_P/GO.

The ideal catalyst should not only have excellent catalytic performance, but also excellent recyclability. Since the prepared N-B-Ni_7_Fe_3_P/SEP and N-B-Ni_7_Fe_3_P/GO have certain magnetic properties, the catalysts in the cuvette could very easily be recycled from the reaction system using strong magnets (Figure 9d–f), and further used to catalyze the hydrogenation reduction of 4-NP to 4-AP to study its recyclability. Figure 8e shows a slight decay in the conversion efficiency of 4-nitrophenol, indicating the high-performance stability of N-B-Ni_7_Fe_3_P/SEP and N-B-Ni_7_Fe_3_P/GO. The FTIR (Figure 1), XRD patterns (Figure 4) and SEM-EDS images (Figure 10) of the catalysts before and after the catalytic reaction also showed no significant changes, indicating the high structural stability of the prepared N-B-Ni_7_Fe_3_P/SEP and N-B-Ni_7_Fe_3_P/GO. In addition, the XPS images show little change in the positions of the binding energy peaks of Ni 2p (Figure 5II(b,d)) and Fe 2p (Figure 5III(b,d)). This indicates that Ni and Fe were not oxidized during the recovery process, suggesting that the catalysts have high compositional stability.

As a result, the remarkable catalytic activity of N-B-Ni_7_Fe_3_P/SEP might be attributed to the following factors: (a) SEP supports with a large specific area and pore volume promote anchored Ni-Fe bimetallic phosphide and 4-nitrophenol; (b) Ni-Fe bimetallic phosphide nanoparticles produced vertically on the carbon carrier surface ensure that Ni-Fe particles can perform catalytic functions; (c) the electron transfer between Ni-Fe bimetallic phosphide and SEP helps to increase the concentration of local electrons and facilitate the absorption of electrons by 4-NP.

## 3. Methods and Materials

### 3.1. Materials

The catalyst was prepared by an equal molar ratio impregnation and hydrothermal method. Poplar wood (Tianjin, China) was pretreated (SEP) (213 °C, 5 min) by steam explosion, as in previous studies [28]. Sigma Aldrich provided analytical-grade sodium borohydride (NaBH_4_), 4-nitrophenol (4-NP), graphene oxide and methylene blue (Shanghai, China). Sinopharm Chemical Reagent Co. Ltd. provided analytical-quality nickel nitrate hexahydrate, iron nitrate hexahydrate and ethanol (Shanghai, China).

### 3.2. Preparation of Catalyst

The detailed preparation process of the catalyst refers to an earlier-reported method [29]. In a typical preparation procedure (Figure 1), 2 g SEP was dispersed into 100 mL deionized water, and then nickel nitrate hexahydrate and ferric nitrate nine hydrate were added according to the molar ratio (Ni:Fe = x:y). The mixture was ultrasonically treated for 15 min before being agitated for 40 min to thoroughly dissolve. After the complete desorption of Ni^2+^ and Fe^3+^, a 5 mL NaBH_4_ (0.5 mol/L) solution was added and rapidly agitated at room temperature for 1 h. Boron-containing metal oxides were grown vertically in situ in SEP. The resultant product was centrifuged and washed with deionized water and ethanol before being dried in a vacuum for 48 h to provide a carbon-based catalyst precursor. The carbon-based catalyst precursor is hereinafter referred to as N-B-Ni_x_Fe_y_/SEP.

Precursors and 200 mg NaH_2_PO_2_ were placed in both end areas of the ceramic. The temperature rose at a rate of 5 °C/min while under nitrogen protection. The N-B-Ni_x_Fe_y_/SEP was maintained at 350 °C for 1.5 h, before allowing it to cool naturally in the furnace to room temperature. The resulting carbon-based catalyst is hereinafter referred to as N-B-Ni_x_Fe_y_P/SEP.

The catalyst preparation process using GO as the carrier was similar to the above, except that the SEP carrier was replaced with GO.

### 3.3. Catalytic Reduction in Catalyst

A Fourier transform infrared (FT-IR) spectrometer (Bruker, Karlsruhe, Germany) using KBr pellet technology was used to measure FT-IR in the wavelength range of from 4000 cm^−1^ to 400 cm^−1^. The morphology and microstructure of the samples were examined by scanning electron microscopy (Zeiss Merlin, Oberkochen, Germany) and transmission electron microscopy (F200X) at an accelerating voltage of 10 kV. Elemental mapping studies were also captured. Thermogravimetric analysis (TGA) was conducted in a thermogravimetric simultaneous thermal analyser (TA, New Castle, DE, USA). The samples of lignin were heated from room temperature to 800 °C at 10 °C/min, in a nitrogen atmosphere. The crystal structures of the samples were examined before and after the catalytic reaction using an Ultima IV X-ray diffractometer (XRD) (Rigaku, Tokyo, Japan) at an operating voltage of 40 kV and a current density of 30 mA in the 2θ range of 10–80°. An ESCALAB 250 analyzer (Thermo Science, Waltham, MA, USA) and a monochromatic Al Ka X-ray source were used for XPS examination. The N_2_ adsorption–desorption isotherm measurements were performed with a BELSORP-mini II instrument (Microtrac, Montgomeryville, PA, USA), using the BET method [30]. A UV-2900 spectrophotometer was used to record the ultraviolet-visible (UV-Vis) absorption spectra (Hitachi, Tokyo, Japan).

### 3.4. Catalytic Ability Test of N-B-Ni_x_Fe_y_P/SEP and N-B-Ni_x_Fe_y_P/GO

The reduction in 4-NP was conducted in a quartz cuvette and monitored by performing UV-Vis spectroscopy (Hitachi UV-2900) at room temperature, according to the reported method [28]. In a typical experiment, a total of 25 μL of the aqueous 4-NP solution (0.01 M) was mixed with 2.5 mL of freshly prepared NaBH_4_ (0.01 M) solution. A set quantity of N-B-Ni_x_Fe_y_P/SEP or N-B-Ni_x_Fe_y_P/GO was then added to initiate the catalytic hydrogenation reaction. UV spectroscopy was used to monitor the reduction in situ, by measuring the absorbance of the solution at 400 nm over time. After the catalytic reaction, the sample can be recovered by strong magnets, due to its strong magnetic properties, and the secondary catalytic reaction can then be performed, according to the aforementioned experimental procedure.

The same procedure was repeatedly used to study the catalytic degradation of MB.

## 4. Conclusions

In summary, using economical and recyclable fibre raw materials as carriers, nickel-supported catalysts were prepared by adsorption and reduction at room temperature. They possess a higher catalytic activity and sustained reusability than graphene oxide-supported catalysts. Their mechanism may have the following aspects: (1) A larger pore size is favourable for the mass transportation of reactant to the inner reaction site of the catalyst. A higher surface area would improve 4-NP and MB adsorption and, consequently, reaction rates. The vertical growth of bimetallic phosphide on SEP/GO ensures the formation of ultra-small particles with high reactivity; (2) The interactions between Ni, Fe and P synergistically enhance the catalytic performance; (3) The presence of NiFeP nanocrystals, as well as their strong contact with SEP/GO, can boost charge transport and the kinetics of interfacial electron transfer; (4) The strong interfacial interaction between bimetallic phosphide and carbon can enhance the dispersion of the nanoparticles, as well as preventing the nanoparticles from dissolving during catalysis, improving stability, and thereby achieving catalytic durability.

## Data Availability

The data presented in this study are available on request from the corresponding author.

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
