# Peer review of "Facile Construction of Iron/Nickel Phosphide Nanocrystals Anchored on N-B-Doped Carbon-Based Composites with Advanced Catalytic Capacity for 4-Nitrophenol and Methylene Blue"

_ijms, 2022, doi:10.3390/ijms23158408_

Round 1
Reviewer 1 Report
After reading the manuscript, I regard it interesting and worthy of publication in this journal. However, before doing so, I have several remarks to make:
a) Firstly, although the paper has plenty of characterization data, I do not know clearly if the bimetallic particles are an alloy indeed or they are just separated NiP and FeP. This is something I regard important to ascertain, since their performance is rather different.
b) Figure 3: the images related to Fe and Ni are completely black so I couldn´t see anything. Is this correct or it is a mistake? Please, check it.
c) Literature: I think just 24 references is a low number. I think the authors should add more references since there is plenty of literature in this field
Author Response
1. Response to comment: (Firstly, although the paper has plenty of characterization data, I do not know clearly if the bimetallic particles are an alloy indeed or they are just separated NiP and FeP. This is something I regard important to ascertain, since
their performance is rather different.)
Response: We report a simple method for constructing Iron/Nickel phosphides nanocrystals anchored on N-B-doped carbon-based composites: utilising steam-exploded poplar (SEP) and graphene oxide (GO) as a carrier.
2. Response to comment: (Figure 3: the images related to Fe and Ni are completely black so I couldn ́t see anything. Is this correct or it is a mistake? Please, check it.)
Response: In Figure 3, we use different colors to distinguish different elements, such as green for Fe elements and blue for Ni elements, and we can find that the elements
are evenly distributed in the measurement range.
3. Response to comment: (Literature: I think just 24 references is a low number. I think the authors should add more references since there is plenty of literature in this field)
Response: Considering the Reviewer’s suggestion, we have added some references
Reviewer 2 Report
The paper well written although needs some minor language corrections. In some places present tense has been used which should converted into past tense. Abstract should contain more numerical data of findings. At the end of abstract there is need include o ne sentence about concluding remarks one about future perspective. The list of key words must be in alphabetical order. Also omit those words from this list which are part of the manuscript title.
In section 2.1 chemical manufacturer names are included but country or city information are absent. The major flaw in the paper is that the authors have not provided the spectroscopic information about dyes degraded products. Did the author have performed NMR, or Mass analysis. What is base of their claim that this product has been formed.
Reviewer 3 Report
Author reported a method for synthesizing Iron/Nickel phosphides nanocrystals anchored on N-B-doped carbon-based composites by utilising steam-exploded poplar (SEP) and graphene oxide (GO) as a carrier. The stability and catalytic activity of synthesized also explored. However, few points may consider before publication.
The title should be short type.
The language of the whole article must cheched by English native as many mistakes are there.
In figure 10 a and b, a great drop in the peak was shown so may be try to put of 5 sec between 0 and 10.
The concept of the figure 10 d, e and f is not very clear, specially a confusion in 10 e.
The authors needs to put relavent references to each section as only 25 references are mentioned.
New Journal of Chemistry 42 (3), 1995-2005, Biomaterials Science 9, 4854-4869, Reaction Kinetics, Mechanisms and Catalysis, 1-15,Energy and Environment Focus 2 (1), 73-78, Journal of Molecular Structure 1098, 393-399.
Need to find the intermediate product of 4-nitrophenol and methylene blue and analysed by GCMS.
Round 2
Reviewer 2 Report
ok
Reviewer 3 Report
can be accepted.